# A Review of *Salmonella* and *Campylobacter* in Broiler Meat: Emerging Challenges and Food Safety Measures

**DOI:** 10.3390/foods9060776

**Published:** 2020-06-11

**Authors:** Hudson T. Thames, Anuraj Theradiyil Sukumaran

**Affiliations:** Department of Poultry Science, Mississippi State University, Starkville, MS 39762, USA; htt37@msstate.edu

**Keywords:** *Salmonella*, *Campylobacter*, Prevalence, Antimicrobials, chicken, biofilm, bacteriophage

## Abstract

Poultry is one of the largest sources of animal-based protein in the United States. Poultry processing has grown from a small local network of plants to nearly 500 plants nationwide. Two of the most persistent bacteria in poultry processing are *Salmonella* and *Campylobacter*. It was not until the introduction of Hazard Analysis and Critical Control Point systems in 1996 that major efforts to reduce bacterial contamination were developed. Traditionally, chlorine has been the industry standard for decontaminating chicken meat. However, antimicrobials such as peracetic acid, cetylpyridinium chloride, and acidified sodium chlorite have replaced chlorine as primary antimicrobials. Despite current interventions, the emergence of stress-tolerant and biofilm-forming *Salmonella* and *Campylobacter* is of primary concern. In an effort to offset growing tolerance from microbes, novel techniques such as cold plasma treatment, electrostatic spraying, and bacteriophage-based applications have been investigated as alternatives to conventional treatments, while new chemical antimicrobials such as Amplon and sodium ferrate are investigated as well. This review provides an overview of poultry processing in the United States, major microbes in poultry processing, current interventions, emerging issues, and emerging technologies in antimicrobial treatments.

## 1. Introduction

In 1937, the first broiler processing plant in the United States was established in the Delmarva Peninsula of Delaware [1]. Customers were able to buy birds that had the feathers and blood removed. In this first form of processing, birds were sold with the head, feet, and entrails intact [2]. During World War II, production grew significantly due to food shortages and is estimated that around 12 processing plants across the Delmarva Peninsula were processing 300,000 birds per day. It was not until after World War II that the majority of broiler processing shifted to the southern region of the United States [1]. By the 1970s and into the early 1980s, 88% of broiler processors resided in southern United States [3]. This was primarily due to the innovation of vertical integration business strategy in which one company manages all the segments of the entire production chain. This greatly increased the production efficiency and led to overall industry growth such that poultry processing employment doubled [1,4]. In the 50 years since modern broiler processing began, the number of processing plants increased from 12 localized plants in Delmarva, to nearly 500 plants nationwide.

This growth in the market for poultry production has been accompanied by several regulatory advancements in the industry, the first being the Poultry Products Inspection Act of 1957. In an effort to prevent the distribution of substandard products, this federal law required that all broiler meat crossing state lines be inspected by the United States Department of Agriculture (USDA) [5]. Another major advancement in poultry processing has been the introduction of Hazard Analysis and Critical Control Point (HACCP) systems. In 1996, Food Safety Inspection services mandated that processing plants under USDA regulation must have a written protocol following the seven HACCP principles [6]. The critical control points identified in the HACCP plan are utilized as locations/points in processing where control measures are applied to reduce/eliminate microbial hazards [6]. Advancements in antimicrobial treatments have helped processing plants meet these new standards put in place by the USDA and Food Safety Inspection Services (FSIS). Many plants utilize a variety of interventions, including physical measures as well as safe antimicrobial treatments such as peracetic acid, chlorine, or cetylpyridinium chloride during processing [7].

One of the primary causes of foodborne illnesses in the United States is *Salmonella,* a bacterium commonly found in the digestive tract of infected chickens [8]. Every year, roughly 1.35 million cases of salmonellosis, an infection by non-typhoidal *Salmonella*, which commonly causes diarrhea and fever, are reported in the United States [9]. Globally, it is estimated that 93.8 million cases are caused by non-typhoidal *Salmonella* every year [10]. Of these cases, 80.3 million are estimated to be foodborne [11]. Contaminated poultry meat is one of the largest contributors to salmonellosis with some studies suggesting that poultry is associated with 25% of outbreaks caused by foodborne pathogens [12]. The usual routes of contamination of poultry meat with *Salmonella* include, leakage of intestinal contents/feces during processing, contaminated processing equipment, water, and the hands of processing workers. Due to the high rate of cross-contamination during slaughter and processing, there are estimated risk levels for *Salmonella* outbreaks from these steps in broiler production. The likelihood of an outbreak occurring from these steps are estimated to be 12% and 33.5%, respectively [12]. Another major concern is the formation of biofilms by *Salmonella* on various poultry processing equipment and surfaces. *Salmonella* in biofilms are more resistant to antimicrobials used for sanitation and persist in the processing facility for prolonged periods increasing the chances of product contamination and foodborne outbreaks [13].

Another pathogen frequently responsible for foodborne illness is *Campylobacter* [14]. In the United States, *Campylobacter* is responsible for an estimated 1.5 million illnesses each year [15]. Globally, *Campylobacter* is the leading cause of bacterial gastroenteritis with around 96 million cases each year [16,17]. Despite there being 30 years of research on *C. jejuni*, the health burden of Campylobacteriosis, which is known to cause diarrhea, fever, and muscle pains, has not been relieved. In fact, an increase in antibiotic resistance has led to increased difficulty in treatments with approximately 310,000 cases being potentially untreatable in the United States [14]. Poultry meat contaminated as a result of cross-contamination from the intestinal tract of infected birds is the primary mode of transmission for *Campylobacter* and is estimated that chickens are responsible for up to 30% of human Campylobacteriosis cases [18,19]. This is due to chickens possessing optimal conditions to host the bacteria’s growth [20]. Horizontal transmission is rapid and flock prevalence can reach 100% within a few days [21]. A lack of visible symptoms causes difficulty in measures to reduce “within-flock prevalence” thus leading to higher prevalence at slaughter [21]. Thus, poultry processing plants are responsible for preventing contaminated meat from reaching consumers. During poultry processing, physical and chemical interventions such as hot and cold-water washes and antimicrobials are applied to mitigate bacterial contamination on meat surfaces [22]. Quantitative microbial risk assessments validate the effectiveness of interventions such as scalding and chilling during processing by reporting an average of 4-log reductions in *Campylobacter* concentrations from before processing to immediately after carcass chilling [19].

Although industry food safety standards have been raised and regulations have improved, *Salmonella* and *Campylobacter* continue to persist in the poultry processing industry. The fight against the health and economic burdens caused by these bacteria continue every year. As new advancements are developed in the poultry industry, it is important to have updated literature to discuss the food safety concerns in modern poultry processing. Therefore, the objectives of this review are to provide insights into the latest challenges in broiler processing posed by *Salmonella* and *Campylobacter*, and to discuss modern advancements in antimicrobial interventions.

## 2. Poultry Borne Pathogens

In the United States, consumption of contaminated food products is responsible for an estimated 9.4 million foodborne illnesses every year [23]. Many foodborne infections are caused by the consumption of contaminated poultry meat [24]. In fact, by analyzing 1114 foodborne outbreaks recorded in the United States from 1998 to 2012, poultry meat was responsible for 279 (25%) of the outbreaks [23]. Poultry meat is known to be a major source for bacteria such as *Salmonella*, *Campylobacter*, *Escherichia coli*, as well as *Clostridium* [23,24]. However, non-typhoidal *Salmonella* and *Campylobacter* remain the leading pathogens causing ever-growing food safety concerns in poultry processing [24,25].

### 2.1. Salmonella

*Salmonella* is a gram-negative bacterium belonging to the Enterobacteriaceae family [26]. The genus *Salmonella* contains two species; *S*. *enterica* and *S*. *bongori*. However, there are many subspecies and serotypes within each species. The most widely accepted method used to differentiate between types of *Salmonella* is serotyping [8]. A serotype classification is based on the properties of the O-antigen and H antigen(s). This is significant because approximately 1500 different serotypes are identified within the *S*. *enterica* species. In total, there are currently over 2500 serotypes of *Salmonella* as described by the World Health Organization. *Salmonella* persists as a prominent infectious bacterium due to such large diversity within the subspecies. Despite the fact that there are many serotypes, only a fraction is responsible for the majority of foodborne outbreaks. The Centers for Disease Control and Prevention (CDC)’s 2005 annual survey demonstrated that the top 30 most reported serotypes from cases of infection represent 81.5% of all isolates reported during 2005 [27]. Based on CDC surveillance technologies, in 2018 the three most common serotypes reported in human infections were *S.* Enteritidis, *S.* Newport, and *S.* Typhimurium [28].

#### 2.1.1. Prevalence in Broiler Meat

Many of the serovars associated with human infections are frequently found in broiler meats [29]. In the past decade, three of the predominant serotypes found in broiler meat were *S*. Typhimurium, *S*. Enteritidis, and *S*. Heidelberg [10,24,29]. Based on USDA reports, *S.* Typhimurium, *S*. Enteritidis, and *S*. Heidelberg were in the top ten serotypes identified in poultry HACCP testing in 2014 [30]. Throughout the processing line, various steps present risk for cross-contamination. The operations with the highest risk of contamination include scalding, defeathering, and evisceration [31]. Noteworthily, the incidence of contamination increases from pre-chill to post-chill despite the levels of *Salmonella* being decreased [32,33]. Among raw poultry products, the highest level of contamination resides in ground broiler meat at roughly 19% whereas contamination in carcasses of young birds is estimated to be around 3.9% [34]. This would suggest that cross-contamination increases as broiler meat becomes further processed [32,35,36]. In retail, studies have shown contamination to be between 19–24% [37]. In a study conducted in Pennsylvania, 19% of prepackaged broiler meat from retail grocery stores were contaminated with *Salmonella* [37]. The level of contamination in open display, organic, and packages labeled ‘Antibiotic-Free’ was slightly higher at around 23% to 24%. A summary of the reported prevalence levels of *Salmonella* in various broiler products is presented in Table 1.

#### 2.1.2. Illnesses and Outbreak Statistics

In the United States, the incidence of outbreaks is monitored by the Foodborne Disease Active Surveillance Network (FoodNet). In 2018, there were 25,606 foodborne illness cases reported with 9084 being caused by *Salmonella* [48]. Salmonellosis is rarely fatal; however, it can lead to severe gastrointestinal upset, hospitalization, and irritable bowel syndrome [49]. For example, a 29-state outbreak of *S. Heidelberg* from rotisserie chicken products resulted in 634 illnesses and 241 hospitalizations [23]. Moreover, it can be fatal for immunocompromised individuals such as HIV and cancer patients as well as older individuals. Inevitably, the onset of this illness from poultry products is due to mishandling and preparation. In a statistical survey observing 88 poultry-borne outbreaks, the major contributory factors were analyzed. The factors with the highest percentages of contribution were related to the handling of the products at home. Improper refrigeration was involved in 48% of the outbreaks, preparing food one or more days before serving was involved in 34% of the outbreaks, and inadequate cooking was involved in 27% [50,51]. Studies have shown that improper refrigeration, at above 8 °C, can lead to significant *Salmonella* growth (2–3 log increase) within 8 days [50]. The recommended minimum internal temperature for all cooked poultry meat is 165 °F as per the United States Department of Agriculture [52]. However, many consumers do not follow these practices, thus putting themselves at risk of infection—for example, a study conducted by UC Davis with 120 participants determined that 40% of the participants undercooked their chicken, 65% did not wash their hands before handling raw chicken, and 50% of participants washed raw chicken in sinks prior to preparation, which are not recommended practices based on USDA guidelines [53]. The economic impact of *Salmonella* outbreaks affects both consumers as well as businesses. In 1988, it was estimated that the total patient-related cost of salmonellosis was between USD 275 million and 1.1 billion [54]. In 1994, the total economic burden of salmonellosis was estimated to be between USD 1 and 2.3 billion and the estimated cost per case was between 500–1350 USD [51]. In recent years, the estimated cost of *Salmonella* contamination in poultry products is USD 2.8 billion based on a 2011 illness model reported by the Centers for Disease Control and Prevention (CDC) that utilized an updated economic cost model to account for cost variables and assumptions [55].

#### 2.1.3. Performance Standards

Given that the prevalence of *Salmonella* varies between different stages of processing and products that are sold in retail, the USDA has different performance standards for chicken products. The goal of these performance standards is not specifically to determine the prevalence of *Salmonella* on different products, but to monitor the effectiveness of the processing procedures in limiting contamination [56]. For broiler carcasses, comminuted chicken, and chicken parts, the maximum acceptable percentages of those positive for *Salmonella* in a 52-week period are 9.8%, 25%, and 15.4%, respectively [56]. Facilities that run the risk of failing to meet the maximum allowed percent positive during the most recent completed 52-week window receive a warning from USDA-Food Safety Inspection Services (FSIS) and are advised to make immediate changes to avoid failing the performance standard. Facilities that fail to meet the performance standard receive a notice of failure and are required to meet with FSIS personnel to develop corrective measures. FSIS also performs follow-up sampling. The issue is to be resolved within 30 days before further enforcement action is taken [56]. Despite numerous improvements in food safety measures and stricter inspection protocols, there are still emerging challenges associated with *Salmonella* in broiler processing.

#### 2.1.4. Emerging Challenges

As mentioned before, the formation of biofilms is one of these challenges. A biofilm is an accumulation of microbial cells that irreversibly attaches to a surface and is enclosed in a polysaccharide matrix [57]. Biofilms were first observed in the 1960s. However, it was not until the late 1980s and 1990s that the phenotypic properties were studied [58]. Biofilm formation is a multistep process involving attachment to a carrier surface, binding to the surface, the development of microcolonies, and the maturation of the biofilm [59]. Biofilms allow *Salmonella* to survive in unfavorable conditions by attaching to abiotic surfaces, most commonly metal, plastic, or glass surfaces [59]. An extracellular matrix comprised of cellulose, biofilm-associated protein, and curli give biofilms unique abilities to resist antimicrobials and prevent the diffusion of sanitizers that are commonly used in poultry processing plants [58,59]. Once formed, biofilms increase the risk of cross-contamination as *Salmonella* persists on processing surfaces that have significant contact with broiler meat.

Another major concern is the development of environmental stress tolerance in *Salmonella*. This includes heat tolerance, acid resistance, antimicrobial resistance, and sanitizer resistance. However, these mutations appear to be strain or serotype specific. For example, *S.* Senftenberg 775 W has proven to be extremely heat tolerant [60]. *S*. Enterica strains have shown significant variation in heat and acid tolerance with log reductions varying between 0.84 and 5.75 log CFU/mL at 57 °C and a pH of 3.0 as seen in Table 2 [61]. In a multistate outbreak of *S.* Heidelberg from a single poultry processor, it was determined that isolates possessed enhanced heat tolerance and virulence capabilities [60]. Between 2013 and 2014, reoccurring outbreaks of this enhanced *S.* Heidelberg were responsible for 634 infections [60]. Current investigations suggest that the persistence or even influx of certain serotypes of *Salmonella* in poultry is due to plasmids bearing resilient virulence genes that are easily transmissible. [62]. The development of resistance against existing antimicrobials has forced new innovations in antimicrobial protocols which can be seen by the reduction in use of chlorine and increase in use of antimicrobials such as peracetic acid [63].

### 2.2. Campylobacter

The first reported human case of *Campylobacter* was discovered in 1913 by Veterinarians McFaydean and Stockman [72]. However, it was not be named *Campylobacter* until 1973 when Veron and Chatelain reclassified *Vibrio fetus* to *Campylobacter fetus* [72]. All campylobacters share the form of spiral-shaped rods [73]. However, variations in the outer membrane of campylobacters are species specific and give unique phenotypic characteristics to the species [74]. To date, the genus *Campylobacter* is composed of 16 species. The species responsible for the majority of human infections are *C. jejuni* and *C. coli* [18]. Most *Campylobacter* growth is restricted to microaerophilic conditions (an atmosphere comprising of 5% oxygen, 10% carbon dioxide, and 85% nitrogen) and within a temperature range of 31–44 °C (optimum 42 °C; [73,75]). Therefore, thermophilic species such as *C. jejuni* and *C. coli* are most persistent in flocks where this environment is readily available [76]. *C. jejuni* typically lines the gastrointestinal tract of chickens where the microaerophilic conditions are optimal by colonizing the mucus of epithelial cells in the ceca and small intestine [21]. It can metabolize mucin and is well adapted to the gastrointestinal tract where mucus film is readily available and atmospheric conditions are met. Extensive research has indicated that poultry is one of the major modes of transmission for this bacterium [77]. Campylobacters in poultry farms are most commonly ingested by the birds during the growing phase [77]. Once ingested, Campylobacters rapidly colonize the gastrointestinal tract, most frequently in the ceca, large intestine, jejunum, and cloaca [73,77]. From there the bacteria can spread throughout the flock. Despite many interventions, human *Campylobacter* infections occur every year from contaminated poultry meat.

#### 2.2.1. Prevalence in Broiler Meat

According to reports from the Centers for Disease Control and Prevention, there are nearly 1.5 million cases of *Campylobacter* infections each year in the United States [15]. It is estimated that 20% to 30% of these are related to poultry meat consumption [78]. Despite developments in antimicrobial treatments, C. *jejuni* is frequently detected in retail broiler products [66]. Table 1 summarizes the prevalence of *Campylobacter* in various broiler products. Retail products with the highest prevalence of *Campylobacter* are whole chicken carcasses [79]. However, *Campylobacter* can be found in other products such as chicken breasts. In a study analyzing 6138 chicken breast samples collected from various retail stores between 2002–2007, 49.9% tested positive for *Campylobacter* [42]. Similarly, in a study with whole carcasses, 43.3% tested positive for *Campylobacter* [43]. In 2013, another study reported the *Campylobacter* contamination of a variety of chicken products from retail grocery stores to be around 38% [44]. There is a considerable amount of evidence suggesting cross-contamination of poultry meat plays a pivotal role in reported human infections [80,81]. In broiler processing, the risk of cross-contamination increases through processing steps where feces and digesta can contact broiler meat products [79]. In fact, during steps such as evisceration, spilling 5 mg of cecal content containing *Campylobacter* has been shown to increase concentrations by 0.6 log CFU in prechill rinses [79]. Cross-contamination of broiler meat can also occur at the home of the consumer. The refrigeration of meat can allow the bacteria to spread to other meats stored in the same packaging. Studies have shown that, at normal refrigeration temperatures of 4 °C, *Campylobacter* can survive for up to 18 days without demonstrating any reduction in bacterial counts [50,82,83].

#### 2.2.2. Illnesses and Outbreak Statistics

Human infections are caused most often through the ingestion of undercooked poultry meat [18]. As stated before, the estimated incidences of infection from *Campylobacter* in 2018 was 1.5 million [15]. This is an increase from the previous year’s reports of nearly 1.3 million cases [84]. Between 2004 and 2012 the number of cases nearly doubled, with the total number of confirmed outbreaks reaching 347 [84]. Based on CDC data libraries, *Campylobacter* outbreaks represent around 1.9% of all foodborne outbreaks every year in the United States [85]. Although rarely fatal, persons infected become at risk of other medical complications such as irritable bowel syndrome and Guillain-Barre Syndrome [84]. The immune response from *Campylobacter* infections has also led to a surge in reports of reactive arthritis [66], suggesting that infections can lead to sudden inflammation of joints [86]. However, the estimated incidence level of reactive arthritis from *Campylobacter* infections is relatively low at 9 out of 1000 [86]. The CDC utilizes 4 major surveillance systems including Nationally Notifiable Disease Surveillance System, National Outbreak Reporting System, National Antimicrobial Resistance Monitoring System, and Foodborne Diseases Active Surveillance Network to complete a comprehensive view of the impact of foodborne infections. Analysis of the data from these networks show that the majority of Campylobacteriosis cases occur in the metropolitan areas of the United States with one third of annual cases originating in Western United States [84]. The average cost of non-hospitalized cases in the United States is nearly 2000 USD and that of hospitalized cases is a staggering 8915 USD [87]. By utilizing an enhanced model to compare estimated costs of various illnesses, the estimated economic burden of *Campylobacter* in the United States is between USD 1.3 billion and 20 billion annually based on CDC statistics [88].

#### 2.2.3. Performance Standards

Similar to *Salmonella*, the prevalence of *Campylobacter* varies between different stages of processing and products that are sold in retail. So, the USDA has different performance standards for the various broiler processing steps. The methodology for the performance standards has been revised throughout the last 5 years. The current method involves collecting samples weekly while keeping track of the percentage of samples positive for *Campylobacter* in a moving 52-week window [56]. This approach allows the FSIS to more accurately determine the processing plant’s performance on a continuous basis [56]. As stated previously the primary purpose of the performance standards tests is to monitor the effectiveness of the processing procedures in limiting contamination [56]. For broiler carcasses, comminuted chicken, and chicken parts, the maximum acceptable % positive for *Campylobacter* in a 52-week period is 15.7%, 1.9%, and 7.7%, respectively [56]. When a processing plant is at risk of failing to meet the performance standards during the most recent completed 52-week window, FSIS notify the processing plant with a warning advising immediate changes to avoid failing the performance standard. In the case that a facility does fail to meet the performance standard, a notice of failure is sent, and the plant representatives are required to meet with FSIS personnel to develop corrective measures. FSIS will perform follow-up sampling throughout the corrective procedures. The plant has 30 days to resolve the issue before further action is taken by FSIS [56].

#### 2.2.4. Emerging Challenges

Recent studies have investigated biofilm formation of various species of Campylobacter. It is well established that many *Campylobacter* species are thermophilic and require microaerophilic conditions to thrive. However, the prevalence of Campylobacter continues to grow in broiler production and human infections. It is believed that strains of Campylobacter are able to attach to various surfaces such as glass and form bacterial pellets for biofilm formation [89]. A biofilm-forming ability has been inconsistently observed among isolates collected from contaminated broilers with only seven out of 60 isolates demonstrating the ability to form biofilms in one study, as seen in Table 4 [67]. A particularly concerning aspect of the biofilm formation of *Campylobacter* is its heightened survivability at low temperatures [90]. One study observed the prolonged survival of *C. jejuni* at temperatures as low as 4 °C [91]. Implications from these results may suggest certain *Campylobacter* strains are capable of developing increased tolerances to temperature stress. Another major concern is the effect of horizontal gene transfer during biofilm formation of *Campylobacter*. The transfer of antibiotic-resistant genes is specifically concerning for human infections where severe infections are treated with antibiotics. Recently, it was observed that *C. jejuni* is highly capable of adapting chromosomally encoded antibiotic-resistant genes through biofilm formation [92]. The consumption of chicken meat contaminated with antibiotic-resistant *Campylobacter* could contribute to the increase in reported infections in the United States despite interventions in place.

Emerging issues with planktonic *Campylobacter* have been observed as well. Despite requiring microaerophilic and thermophilic conditions for survival, *Campylobacter* has been noted to exhibit extreme stress tolerance [68]. Genome sequencing has shown that many *Campylobacter* strains lack common stress response genes [68]. However, multiple studies have indicated that challenging *C. jejuni* under high stress conditions enhanced tolerance to multiple factors such as heat and acids [69,93]. In cases involving human infections, there is a high prevalence of aerotolerant *Campylobacter* strains. One recent study found that out of 121 sample strains, 65 were considered hyper-tolerant and 46 were tolerant to oxygen [70]. Strains hyper-tolerant to oxygen also exhibited tolerance to other stressors. Of the 65 strains hyper-tolerant to oxygen, 56 were tolerant to heat treatment at 72 °C for 30 s [70]. Because *Campylobacter* has been associated with high environmental sensitivity, the survivability of *C. jejuni* has been underestimated, which may suggest increased monitoring of this pathogen may be necessary [70].

## 3. Current Post-Harvest Antimicrobial Interventions

The 2005 ban on the use of antibiotics in chicken feeds by the FDA has led to several advancements in antimicrobial techniques used during broiler processing in the United States. This has put greater emphasis on post-harvest strategies to decrease microbial prevalence. The application of chemical antimicrobials is the most common intervention in poultry processing today [67]. Traditionally, chlorine was the industry standard as an antimicrobial treatment [67]. However, compounds such as peracetic acid (PAA), cetylpyridinium chloride (CPC), acidified sodium chlorite (ASC), and trisodium phosphate (TSP) have become more common as they exhibit greater microbial reduction [67,94,95]. After early processing stages such as plucking and evisceration, antimicrobials are usually applied as sprays or dips at various stages of broiler processing. The efficacies of these treatments from various studies can be found in Table 3.

### 3.1. Peracetic Acid

In recent years, one of the most commonly used antimicrobials is peracetic acid (PAA). It has been approved by the U.S. Food and Drug Administration and is regarded as a processing aid by the USDA-FSIS [48]. It can be applied as a spray, dip, or rinse in chilling tanks. The maximum permissible limit for PAA is 2000 ppm as per the USDA standards [115]. In order to prevent color or flavor changes in meat products, the compounds utilized to form peracetic acid are regulated. The maximum mixture of acetic acid and hydrogen peroxide as a peracetic acid treatment is 0.022% and 0.012% respectively [116]. Although used in different stages of processing, PAA is most effective in chilling applications of carcasses as per various studies [67,98,117]. This is due to PAA being highly soluble in water allowing for a more complete coverage on chicken carcasses in chilling tanks [48]. Studies on the efficacy of PAA in carcass chilling applications have noted significant reductions in *Salmonella* and *Campylobacter* with reductions reaching 91% and 43%, respectively, with concentrations as low as 85 ppm despite the maximum permissible limits being much higher [116]. Using concentrations of PAA at 0.02% (200 ppm) has shown the most consistent success with reductions in *Campylobacter* in carcasses reaching 1.5 log CFU [116]. Peracetic acid has also shown great success in the post-chill dip treatment of chicken breasts. A recent study noted 4.08 and 2.23 log CFU/chicken reductions in *Campylobacter* and *Salmonella,* respectively in chicken breasts exposed to a 15 s dip in 750 ppm PAA [48]. It has also been found that PAA treatments on ground chicken were more successful at reducing *Salmonella* and *Campylobacter* than other antimicrobials. One study reported 0.1% PAA solution reduced *Salmonella* and *Campylobacter* on ground chicken by 1.5 logs and 1.3 logs, respectively [96]. Similarly, [97] reported a 1.4-log reduction in *Salmonella* on ground chicken frames dipped in 0.1% PAA. Further research on the efficacy of peracetic acid continues and current investigations look to enhance the bactericidal capabilities of this antimicrobial in conjunction with other compounds.

### 3.2. Cetylpyridinium Chloride

Another antimicrobial utilized in poultry processing is cetylpyridinium chloride (CPC). CPC inhibits bacterial growth by interacting with the acid groups of bacteria creating ionized compounds which prevent bacterial metabolism [118]. It is a considered a strong antimicrobial and highly regulated with the current maximum permissible limit being 0.3 g per pound of raw poultry carcasses as a spray treatment, and 0.8% by weight of carcasses as a liquid aqueous solution [119]. According to the current USDA regulations, CPC utilized as a dipping application cannot have more than 10 s of contact with broiler meat [94]. However, exposing broiler carcasses to CPC in post-chilling tank simulations has demonstrated significant log reductions at exposure times of greater than 10 s. Treating drumsticks with 0.6% CPC for 30 s resulted in a 4-log reduction in *Salmonella* whereas exposure for 10 s resulted in a 3-log reduction [94]. Results suggest that the efficacy of CPC is significantly greater at exposure times above the current USDA regulations [94]. Many studies have reported success in reducing bacterial counts with CPC as a spray though not as significant as immersion with log reductions in bacteria averaging between 1.5–2.9 [120]. Spraying ground chicken meat with a 0.5% solution of CPC resulted in only 0.5-log reduction in *Salmonella* [97]. Similarly, spraying ground chicken with a 0.6% CPC solution resulted in approximately 0.8-log reductions in *Salmonella* and *Campylobacter* counts [96]. With current restrictions on the utilization of CPC in immersion, advancements in spray mechanisms may find greater use of CPC in post-chilling interventions.

### 3.3. Acidified Sodium Chlorite

Another antimicrobial that has been approved by the USDA for use in poultry processing is acidified sodium chlorite (ASC) [121]. The maximum permissible limit of ASC as a spray or dip treatment is 1200 ppm [122]. However, in chilling applications, the maximum permissible limit is 150 ppm [122]. Acidified sodium chlorite acts by disrupting the cellular membrane of bacteria and oxidizing the constituents of the microorganisms [122]. Kemp et al. (2000) reported that dipping carcasses for 5 s in a 1200 ppm solution of ASC resulted in significant reductions in multiple bacteria. Aerobes, *E. coli*, and coliforms were reduced by 1.03, 2.31, and 1.96 logs, respectively [121]. Purnell et al. (2013) found that chicken breasts and chicken necks treated for 30 s with ASC as a spray at 1000 ppm resulted in 1.28-log and 1.60-log reductions in *Campylobacter* [102]. Acidified sodium chlorite performed better than peracetic acid on chicken breasts at both 30 s and 15 s of contact resulting in a difference of 0.13- and 0.04-log reductions, respectively [102]. Acidified sodium chlorite has also proven to be an effective treatment against *Salmonella*. Dipping chicken drumsticks and chicken breasts in a 1200 ppm ASC solution for 1 min resulted in 1.8-log and 0.9-log reductions in *Salmonella* [101]. Although effective in reducing microbial loads on poultry, ASC has demonstrated deleterious effects on sensory characteristics of poultry meat. Nagel et al., 2013, reported that multiple studies had acknowledged color and odors changes on broiler meat treated with 1200 ppm of ASC [98]. However, most of the deleterious effects disappeared during water chilling of poultry products [98].

### 3.4. Chlorine

Traditionally, chlorine has been the leading antimicrobial in broiler processing, most commonly utilized in chiller applications [63]. Studies before the mid-2000s noticed reductions in bacterial counts as high as 4 logs [120]. Others saw reductions nearing 60–70%. However, in recent years, the effectiveness of chlorine has become inconsistent. In a study investigating the efficacy of chlorine in chilling applications of carcasses, 30 ppm of chlorine reduced the percentage of positive isolates by 56.8% and 12.8% for *Salmonella* and *Campylobacter,* respectively [63]. On the other hand, an 85-ppm mixture of PAA and hydrogen peroxide resulted a 91.8% and 43.4% reduction [63]. Studies have suggested for years that in order to kill bacteria such as *Salmonella* in broiler skin, concentrations of chlorine need to be significantly higher. One study found a difficulty in killing *Salmonella* until concentrations as high as 400 ppm and 800 ppm were reached [123]. Trends in data from various studies would suggest common bacteria found in broiler processing have developed a greater tolerance to chlorine-based antimicrobials. One study noted that treating broiler drumettes with 50 ppm of chlorine in chilling applications had the same effect as using distilled water. In both instances on drumettes, 2 to 3 log CFU/mm of *Salmonella* and *Campylobacter* were recovered [103]. Even more concerning are the correlations between chlorine resistance and multidrug-resistant serovars of *Salmonella* [124]. Unfortunately, increasing concentrations of chlorine is not an option. In order to comply with the demand for cleaner food and less use of chemicals in processing, the maximum permissible limit for chlorine is now 50 ppm [115]. If chlorine is used in processing plants, it is typically not used independently. Many studies have found greater success in reducing bacterial counts in chilling applications utilizing chlorine dioxide simultaneously with chlorine [125]. One of the major limitations of using chlorine in chilling applications is its sensitivity to the environment. Chlorine’s efficacy is greatly reduced by a high pH (>7) and large quantities of organic content [63]. This organic content appears in the foam that forms on chilling tank surfaces and must be scooped off. While still useful under specific contexts, chlorine is proving to be less effective and is less common in broiler processing. Chlorine use in poultry processing is also banned in European countries and Russia. This, in turn, presses export restrictions on US processing plants.

### 3.5. Trisodium Phosphate

Another antimicrobial commonly used in conjunction with other compounds is trisodium phosphate (TSP). Trisodium phosphate is allowed as a spray or dip application with a maximum permissible limit of 12% in a mixed solution as per the USDA regulations [115]. This antimicrobial has unique characteristics that create a high pH between 11.2 and 12.1 on broiler meat surfaces [126]. It is thought that the effects of the higher pH assist in hindering major aerobic bacteria. Some studies have noted greater efficacy against gram negative bacteria specifically [127]. In many studies, treating chicken with trisodium phosphate typically results in a 1.5- to 3-log reduction in bacterial counts [104]. One reported controversial effect of trisodium phosphate treatments is the accelerated spoilage development during storage on chicken treated with TSP [104]. However, other studies refute these claims. By multiple comparisons of bacterial growth on meat treated with and without TSP, Rio et al., 2007, argued that TSP did not accelerate spoilage bacteria growth [128]. However, bactericidal effects of TSP decreased during refrigerated storage. Rio et al., 2007, inferred that gram-positive bacteria treated in the study may be more tolerant to TSP due to differences in the cellular membrane of the bacteria [128]. Gram-positive bacteria such as *Listeria monocytogenes* which can survive in low temperatures and in high concentrations of TSP could become more problematic as a food safety concern for poultry in storage.

## 4. Emerging Technologies/Interventions

Currently, major investigations are being conducted to improve intervention methods in poultry processing. Processing plants cannot rely on the integrity of cold transportation of food products to retailers and therefore, must prioritize advancements on antimicrobial interventions. There are many factors driving industrial changes but one of the most prominent factors is pressure by public demand for safer mechanisms [129]. The factors influencing the acceptability of microbial interventions are typically the level of concern people associate with the intervention, public awareness of an intervention, and severity of consequences if the intervention was not in place [129]. There are currently many chemical interventions utilized in poultry processing. As stated previously, the major compounds include PAA, chlorine, CPC, and TSP. Though effective in their current use, the effect of these compounds is very limited due to current maximum permissible limits in place to prevent chemical residues which can cause harmful effects to consumers or damage the quality of the meat products via discoloration or off-flavoring [67]. Table 4 highlights the recent innovative antimicrobial treatments.

### 4.1. Amplon

One antimicrobial advancement under investigation is the commercial product Amplon. Amplon is comprised of sulfuric acid and sodium sulfate and is approved by the USDA-FSIS for spray, wash, and dip applications [67]. Amplon has proven to be successful in reducing the bacterial counts of *Salmonella* on retail meat products such as chicken wings, where log reductions in bacteria exceeded that of commonly used antimicrobial CPC [135]. The immersion of chicken wings in Amplon for 20 s resulted in a 1.2-log reduction in *Salmonella* [135]. In another study, dipping whole carcasses in Amplon for 15 s post-chilling resulted in a 1.53 log CFU/chicken decrease in *Campylobacter* [67]. Even though limited research on this antimicrobial is available, results from commercial plant simulations do indicate that Amplon could be a promising alternative to the current antimicrobials utilized [67].

### 4.2. Sodium Ferrate

A compound under recent investigation is sodium ferrate. Ferrate is a strong oxidizing agent capable of removing organic and inorganic toxins [136]. Many studies have investigated the efficacy of sodium ferrate as a waste-water treatment and alternative to chlorinating water. Maghraoui et al. (2013) studied sodium ferrate’s biocidal capabilities in water contaminated with *Salmonella* and various other pathogens [130]. Only 8 mg/L of sodium ferrate in water was required to inactivate *Salmonella* whereas inactivation of *E. coli* and *Pseudomonas* required 5 mg/L and 6.3 mg/L, respectively. The study also found that the required dose for consistent bacterial inactivation above 99.9% was 30 mg/L. Sanitizing water with sodium ferrate can significantly reduce costs. One study reported that to disinfect one liter of water requires 4.5 g of chlorine whereas only 0.5 g of sodium ferrate would be needed [137]. Currently only one study has investigated the effects of sodium ferrate as an antimicrobial treatment on broiler meat. Electrostatic spray application of 0.15% and 0.3% solutions of sodium ferrate on chicken thighs reduced *Salmonella* counts by 0.65 and 0.89 log CFU/g, respectively [109]. The technology for utilizing sodium ferrate in machinery has been patented [138].

### 4.3. Electrostatic Spraying

New antimicrobials are not the only aspects of microbial intervention being investigated. New methods of application of these antimicrobials are being thoroughly researched as well. The conventional application of antimicrobial treatments as a dip or spray requires significant amounts of water. Recent studies have investigated the application of electrostatic spraying in antimicrobial treatments of broiler meat [131]. Conversely, electrostatic spraying is not a new invention. Articles dating back to 1964 discuss the design and methods of application of electrostatic spraying for paint products [139]. Electrostatic sprayers operate by applying a negative charge to the compound being sprayed which are attracted to the positively charged surface of broiler products. This, in turn, helps tremendously with accuracy and efficiency of application. Products are also able to have a more even and complete coverage on the surfaces they are applied to. Shen et al. (2019) reported that significantly less water was needed for electrostatic spray applications with approximately 58.2 mL/min being required as compared to an average of 500 L for immersion applications. However, no differences were found in the efficacy in *Campylobacter* reduction. Application of 0.1% PAA as conventional immersion and electrostatic spray on chicken wings resulted in an average reduction of 2.3 log CFU/g and 2.1 log CFU/g of *Campylobacter,* respectively [131]. Another study found similar results between applying peracetic acid at 2000 ppm through immersion and electrostatic spray on chicken wings with applications resulting in 1.82- and 1.9-log reductions in *Salmonella,* respectively [132]. Applications in beef processing have also shown similar results. Hudson (2015) noted that there were no differences between conventional spraying and electrostatic spraying of peracetic acid on beef brisket with reductions in *Escherichia coli* being around 1.0 log and 0.7 log CFU/cm^2^ [140]. Currently more research is needed to determine if electrostatic spray mechanisms effectively increase the efficacy of applied antimicrobials.

### 4.4. Cold Plasma Treatment

In recent years, studies have been investigating the use of cold plasma treatments on food products as an antimicrobial intervention. Cold plasma or nonthermal plasma (NTP) is emitted as gas or through microwaves at ambient temperatures between 30 °C–60 °C by ionizing a controlled atmosphere with electrodes. Nonthermal plasma acts by rupturing the cell membranes of bacteria with electrostatic tension. Ionized oxygen as NTP also has strong oxidizing properties, which easily disrupt bonds within unsaturated fatty acids and the proteins of microbes [141,142]. The utilization of NTP treatments has proven to reduce surface bacteria at costs lower than conventional interventions [143]. Misra and Jo noted, in one study, that a NTP treatment comprised of He/O_2_ successfully reduced aerobic bacterial counts on bacon by 4.5 log CFU/g [141]. In a treatment on pork-butt, NTP effectively reduced *S.* Typhimurium by 2.68 log CFU/g [142]. In an experiment with chicken breast, NTP reduced *Salmonella* counts by 2.71 log CFU/g [110]. Nonthermal plasma can also be applied on in-package products. In modified atmospheres, NTP effectively inhibits microbiological growth which may enhance the efficacy of interventions in prepackaged meats [141]. One study has reported that the application of NTP in packaged chicken breast could extend the shelf life of the product to up to 14 days [141]. Though not yet investigated in many commercial settings, there is high potential for NTP in commercial processing plants.

### 4.5. Bacteriophages

Lastly, recent studies have been further investigating the use of bacteriophages in broiler processing as antimicrobial treatments [122,123]. Despite many chemical antimicrobial agents being used during processing, a tremendous number of outbreaks still occur each year in the United States. Chemical antimicrobials are certainly improving in efforts to counteract developing tolerances common bacteria have acquired. However, current compounds such as PAA, TSP, chlorine, and CPC, kill microbes indiscriminately [144]. This includes beneficial flora which contribute to benefitting shelf life and other sensory attributes of meat products. The use of bacteriophages as antimicrobial treatments could be widely adapted in commercial processing plants as the consumer acceptance of current methods continues to decline [144]. Some commercial bacteriophage preparations such as Listex P100 are already approved by the U.S. Food and Drug Administration as well as European countries [111]. However, many phages have been isolated and studied. Phage cocktails such as Felix-O1, SCPLX-1, and SJ2 have been shown to be effective in reducing *Salmonella* counts on various products [144]. Bacteriophages are bacterial viruses that can infect and kill specifically targeted bacteria [111]. As seen in Table 2, one study has found success in the reduction in *S*. Enteritidis below detectable limits on chicken breasts [111]. Contaminated chicken breasts treated with a bacteriophage mixture saw *Salmonella* reductions around 3.12 log CFU/mL in a mixed suspension [111]. This would suggest that specific bacteriophages can be used to target specific bacterial reduction on broiler products. Another study found that by immersing carcasses in a phage suspension, *Salmonella* counts on chicken skin were reduced by 1 log CFU/cm^2^ [112]. Duc et al. (2018) also noted reductions in *Salmonella* ranging from 1.41–1.86 log CFU/piece on chicken breasts treated with 30 µL of a bacteriophage cocktail [113]. Sukumaran et al. (2015) reported that bacteriophages can be applied sequentially with chemical antimicrobials such as PAA and CPC to achieve *Salmonella* reductions up to 2.3 log CFU on chicken skin [145]. Another study by the same group observed a synergistic effect of bacteriophages with modified atmosphere packaging in reducing *Salmonella* on chicken breast fillets [146]. A small variety of commercial bacteriophage products have been approved by the USDA, but few are used in the industry. Currently, most bacteriophage research is conducted in a laboratory setting. For example, chicken thighs treated with a phage cocktail in a tumbler resulted in an average of 1.1 log CFU/g reductions in *Salmonella* [114]. However, further research is needed to test the efficacy of bacteriophages in commercial processing plants. One major concern is the possibility of phage-resistant bacteria. There have been reports of strains of *S.* Enteritidis developing phage-resistant mutations. However, predictive modeling may provide accurate predictions of the emergence of phage-resistant strains of bacteria [147]. Phage resistance becomes more concerning during biofilm formation. While phage-resistant cells are rare, it has been observed that these cells are able to replicate faster than the bacteriophage can infect the biofilm [119]. Phage-resistant cells will also envelop phage-susceptible cells providing a barrier to protect the biofilm from infection [119].

## 5. Conclusions

The poultry industry is an important sector of agriculture that has dramatically grown in the last 50 years. From small processing plants in Delaware, poultry processing has grown into a multibillion-dollar industry with production surpassing every other meat industry in the United States. Despite the many advancements that have occurred throughout the years, *Salmonella* and *Campylobacter* continue to persist as the prominent pathogens in poultry meat causing foodborne illnesses and outbreaks. These bacteria are constantly monitored by the USDA-FSIS through performance standard protocols for commercial poultry processing plants.

There are several antimicrobial interventions utilized by commercial processing plants to ensure that performance standards are met as well as to ensure the safety of food products distributed to retailors. Chemical antimicrobials are the leading interventions used currently in the industry with peracetic acid, cetylpyridinium chloride, acidified sodium chlorite, chlorine, and trisodium phosphate being the most prominent ones. Despite having an arsenal of interventions available for poultry processing, new food safety challenges are presented by *Salmonella and Campylobacter*. These include formation of biofilms and development of stress tolerance such as acid tolerance and heat tolerance in *Salmonella* as well as aerotolerance/oxygen tolerance in *Campylobacter*. In order to overcome these emerging challenges poultry industry is constantly looking for novel interventions.

Novel chemical antimicrobials such as Amplon and sodium ferrate have shown promising results in reducing *Salmonella* and *Campylobacter* in broiler meat. The electrostatic spraying of antimicrobials is a recent advancement that is being intensely investigated which provides more effective antimicrobial coverage on meat with a much lower utilization of water. The financial savings from a reduced use of the compounds and water involved in these treatments could prove to greatly improve the economical aspect of poultry processing. Though never used in commercial settings, the application of cold plasma technology is also a great emerging technique that can even be applied on pre-packaged foods. The use of bacteriophages is also currently under investigation for the treatment of broiler meat, which has the advantage of targeting bacteria specificity. Although further research is necessary to find optimal applications of this intervention, the current findings demonstrate the value of utilizing bacteriophages as an alternative to conventional measures.

Many advancements have taken place over the years in poultry processing and we are still searching for ways to decrease bacterial counts in poultry meat. With the technologies currently under investigation, conventional measures may soon undergo significant changes. Despite challenges that continue to arise surrounding the food safety of poultry meat, notable technological improvements are still being discovered.

## Figures and Tables

**Table 1 foods-09-00776-t001:** Reports on the prevalence of *Salmonella* and *Campylobacter* in various retail poultry products in the United States.

Bacteria	Name of Product	Level of Prevalence	Year of Sampling	Reference
*Salmonella*	Retail Chicken	84//378 (22%)	2006–2007	[37]
*Salmonella*	Retail Chicken	9/212 (4.2%)	2000	[38]
*Salmonella*	Retail Chicken	158/1320 (12%)	2011	[39]
*Salmonella*	Chicken Breast	1345/10097 (13.3%)	2000–2010	[39]
*Salmonella*	Chicken Breast	60/210 (28.5%)	2000–2010	[39]
*Salmonella*	Chicken Breast	47/105 (44.7%)	2014–2015	[40]
*Salmonella*	Drums Stick	43/105 (41%)	2014–2015	[40]
*Salmonella*	Ground Chicken	13/49 (26%)	2009	[41]
*Campylobacter*	Chicken Breast	3064/6138 (49.9%)	2002–2007	[42]
*Campylobacter*	Whole carcass	84/194 (43.3%)	2006–2007	[43]
*Campylobacter*	Retail Chicken	42/156 (26.9%)	2010	[44]
*Campylobacter*	Retail Chicken	130/174 (70.6%)	2000	[38]
*Campylobacter*	Chicken Breast	4659/9968 (46.7%)	2000–2010	[39]
*Campylobacter*	Boneless meat	308/755 (41%)	2005–2011	[45]
*Campylobacter*	Chicken Breast	584/2376 (24.6%)	2015	[46]
*Campylobacter*	Chicken liver	29/45 (64.4%)	2018	[47]

**Table 2 foods-09-00776-t002:** Emerging food safety issues associated with *Salmonella* and *Campylobacter* in poultry processing.

Bacteria	Issue	Findings	Source
*Salmonella*	Biofilm	Increased biofilm forming ability in *Salmonella*	[59]
*Salmonella*	Heat tolerance	*S.* Senftenberg 775 W extreme tolerance to heat	[60]
*Salmonella*	Acid tolerance	*S.* Enterica Increased tolerance to low pH (3.0)	[61]
*Salmonella*	Antibiotic resistance	*S.* Typhimurium exhibited increased ciprofloxacin tolerance	[64]
*Salmonella*	Cross-tolerance	Exposure to single stressor resulting in increased tolerance to multiple stressors	[65]
*Salmonella*	Thermal processing tolerance	*S.* Typhimurium exhibited high levels of tolerance to dry heat	[65]
*Campylobacter*	Reactive Arthritis	Human infections leading to reactive arthritis	[66]
*Campylobacter*	Biofilm	Increased tolerance to antimicrobials	[67]
*Campylobacter*	Stress tolerance	Increased tolerance to heat and oxygen	[68]
*Campylobacter*	Heat tolerance	Increased tolerance to heat	[69]
*Campylobacter*	Aerotolerance	Increased tolerance to oxygen	[70]
*Campylobacter*	Heat tolerance	Increased tolerance to high temperatures	[70]
*Campylobacter*	Guillain–Barré Syndrome	Chickens infected with *C. jejuni* developed GBS-like paralytic neuropathy	[71]

**Table 3 foods-09-00776-t003:** Efficacy of current and emerging antimicrobial technologies in reducing *Salmonella* and *Campylobacter* in various poultry meat products.

Antimicrobial	Chicken Product	Bacteria	Type of Treatment	Result (Log CFU Reduction)	Reference
PAA (1000 ppm)	Chicken breast	*Salmonella*	Immersion	1.92	[48]
PAA (750 ppm)	Chicken Breast	*Salmonella*	Immersion	2.23	[48]
PAA (0.1%)	Drumsticks	*Salmonella*	Dip	2.0	[94]
PAA (0.1%)	Ground Chicken	*Salmonella*	Immersion	1.5	[95]
PAA (0.1%)	Ground Chicken	*Salmonella*	Immersion	1.4	[96]
PAA (750 ppm)	Chicken Breast	*Campylobacter*	Immersion	4.08	[48]
PAA (1000 ppm)	Chicken Breast	*Campylobacter*	Immersion	1.87	[48]
PAA (0.1%)	Whole Carcass	*Campylobacter*	Dip	2.0	[97]
PAA (0.1%)	Ground Chicken	*Campylobacter*	Immersion	1.3	[95]
PAA (1200 ppm)	Drumsticks	*Campylobacter*	Spray	1.14	[98,99]
CPC (0.6%)	Drumsticks	*Salmonella*	Spray	4.0	[94]
CPC (0.6%)	Ground Chicken	*Salmonella*	Spray	0.8	[96]
CPC (0.5%)	Ground Chicken	*Salmonella*	Spray	0.5	[97]
CPC (0.6%)	Drumsticks	*Campylobacter*	Spray	0.8	[94]
CPC (0.6%)	Ground Chicken	*Campylobacter*	Spray	0.8	[96]
CPC (0.5%)	Whole Carcass	*Campylobacter*	Spray	1.2	[100]
ASC (1200 ppm)	Drumsticks	*Salmonella*	Dip	1.8	[101]
ASC (1200 ppm)	Chicken Breast	*Salmonella*	Dip	0.9	[101]
ASC (1000 ppm)	Chicken Breast	*Campylobacter*	Spray	1.6	[102]
Chlorine (30 ppm)	Whole Carcass	*Salmonella*	Chiller tank	56.8%	[63]
Chlorine (0.003%)	Whole Carcass	*Salmonella*	Chiller tank	No significant difference	[96]
Chlorine (50 ppm)	Broiler Wing Drumettes	*Salmonella*	Chiller tank	No significant difference	[103]
Chlorine (30 ppm)	Whole Carcass	*Campylobacter*	Chiller tank	12.8%	[63]
Chlorine (0.003%)	Whole Carcass	*Campylobacter*	Chiller tank	No significant difference	[96]
Chlorine (50 ppm)	Broiler Wing Drumettes	*Campylobacter*	Chiller tank	No significant difference	[103]
TSP (10%)	Whole Carcass	*Salmonella*	Dip	2.0	[104]
TSP (10%)	Chicken patty	*Salmonella*	Immersion	1.18	[105]
TSP (12%)	Chicken Breast Skin	*Salmonella*	Spin Chiller	1.4	[106]
TSP (8%)	Whole Carcass	*Salmonella*	Dip	2.27	[107]
TSP (12%)	Chicken Breast Skin	*Campylobacter*	Spin chiller	1.8	[106]
TSP (10%)	Whole Carcass	*Campylobacter*	Dip	2.27	[108]
Amplon (3–4 gpm)	Whole Carcass	*Campylobacter*	Spray	3.25	[67]
Amplon (15 s)	Whole Carcass	*Campylobacter*	Post-Chill Immersion	1.53	[67]
Sodium Ferrate (0.15%)	Chicken Thighs	*Salmonella*	Spray	0.65	[109]
Sodium Ferrate (0.3%)	Chicken Thighs	*Salmonella*	Spray	0.89	[109]
Cold plasma	Chicken Breast	*Salmonella*	Spray	2.71	[110]
Bacteriophage	Chicken Breast	*Salmonella*	Not specified	3.12	[111]
Bacteriophage	Chicken Skin	*Salmonella*	Immersion	1	[112]
Bacteriophage	Chicken Breast	*Salmonella*	Not specified	1.86	[113]
Bacteriophage	Chicken Thighs	*Salmonella*	Pipette	1.1	[114]

**Table 4 foods-09-00776-t004:** Advantages and limitations of current and emerging antimicrobial interventions.

Technology	Advantages	Limitations	Max Permissible Limit	Reference
Chlorine	-Industry standard for many years	-Decreasing efficacy due to chlorine tolerance -Efficacy reduced by high pH and organic load	50 ppm	[63]
Amplon	-Improved antimicrobial activity	-Limited research	Mixture flow between 5–10 gal/min in water	[67]
Cetylpyridinium chloride	-Effective antimicrobial as a spray application	-efficacy limited by current USDA restrictions on maximum permissible limits and contact time	Spray: 0.3 g/lb of raw poultry Liquid aqueous solution: 0.8%	[94]
Acidified Sodium Chlorite	-Viable alternative to current chemical antimicrobials	-Can cause deleterious effects on sensory characteristics	Spray or dip: 1200 ppmImmersion: 150 ppm	[98,123]
Trisodium phosphate	-High microbiocidal capacity for gram negative bacteria	-Accelerated spoilage in retail products -Accelerated gram positive bacterial growth	12% in mixed solution	[128]
Sodium Ferrate	-Strong biocidal capabilities	-Limited studies on meat products	NA	[130]
Electrostatic Spray	-Improved antimicrobial coverage on meat -requires less antimicrobials and water	-Mixed results on efficacy -Not effective with certain compounds	NA	[131] [132]
Bacteriophage	-Target specificity	-Limited research in commercial processing	Applied to achieve a level of 1 × 10^7^–1 × 10^9^ plaque forming units	[111]
Peracetic acid	-Most effective antimicrobial in chilling applications	-Limited activity against biofilms	2000 ppm	[67,133]
Cold Plasma	-Cost effective -microbiocidal	-May affect quality attributes	NA	[134]

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
