# Peer review of "A Review of Salmonella and Campylobacter in Broiler Meat: Emerging Challenges and Food Safety Measures"

_foods, 2020, doi:10.3390/foods9060776_

Round 1

Reviewer 1 Report

The manuscript is a review on the different approaches to controlling Salmonella and Campylobacter.  It uses up to date references and treatments being applied to date.

The review is well written and needs minor edits. 

1.    Several places in the review the word liter is abbreviated “L” or “l”.  Please be consistent in the abbreviations.  Examples are line 223 and line 524.   

2.    Introduction

a.    Line 39:  the first broiler processing plant was located on the Delmarva Peninsula.  It is confusing when a single plant can be located in three states.  Is it a single plant or multiple plants?

3.    Line 354: insert space prior to 72o

4.    Table 4 uses old terminology for Salmonella.The authors do an excellent job throughout the manuscript but needs to update this table. S. typhimurium should read S. Typhimurium as well as other serotypes.

 5.   Please remove the “U” on reference number six or spell out FSIS with the United States Department of Agriculture, Food Safety Inspection Service.

Author Response

Response to reviewer comments

We thank the reviewer for all the suggestions and comments to improve the overall quality and suitability of the manuscript for publication. We have addressed all the comments carefully and have highlighted the changes throughout the manuscript.

Reviewer 1

The manuscript is a review on the different approaches to controlling Salmonella and Campylobacter.  It uses up to date references and treatments being applied to date. The review is well written and needs minor edits.

Comment 1: Several places in the review the word liter is abbreviated “L” or “l”.  Please be consistent in the abbreviations.  Examples are line 223 and line 524.  

Response: Inconsistencies in the use of liter abbreviated as “l” or “L” were addressed to be uniform. Liter has been abbreviated as ‘l’ throughout the manuscript.

Comment 2: Introduction

  1. Line 39: the first broiler processing plant was located on the Delmarva Peninsula.  It is confusing. when a single plant can be located in three states.  Is it a single plant or multiple plants?

Response: The confusion of first sentence in the introduction was rectified. Please see line 48 of the revised manuscript. The Delmarva Peninsula has regions claimed by three neighboring states. However, the first processing plant was under Delaware territory.

Comment 3: Line 354: insert space prior to 72o

Response: A space has been added. Please see the correction in line 342 of the revised manuscript.

Comment 4: Table 4 uses old terminology for Salmonella. The authors do an excellent job throughout the manuscript but needs to update this table. S. typhimurium should read S. Typhimurium as well as other serotypes.

Response: Salmonella serotypes in table 4 have been reformatted to meet updated standards.

Comment 5: Please remove the “U” on reference number six or spell out FSIS with the United States Department of Agriculture, Food Safety Inspection Service.

Response: As suggested, the “U” was removed in reference 6 and FSIS was spelled out as ‘United States Department of Agriculture, Food Safety Inspection Service’.

Reviewer 2 Report

Introduction, Lines 63-64

Please add a sentence to emphasize the use of critical control points (eg they could be used to eliminate microbial hazards to an acceptable level etc).

Introduction, Lines 69-75

Please describe the main routes of meat contamination with Salmonella (eg intestinal tract of chicken and infection from environment (feed, water, human interventions etc).

Please provide a short description of salmonellosis (eg clinical symptoms)

Introduction, Lines 83-95

Same as above for Campylobacter and campylobacteriosis

Section 4.4 Cold Plasma Treatment, Lines 562-564

Please provide more information about the mechanism involved (eg conductive electrodes, gas gap, generation of bioactive particles, UV, charged particles etc)

Finally, the integrity of the cold chain process during product transportation and selling of final product should also be mentioned in this article, as an additional food safety measure (in order to ensure product quality and sustain public health).

Author Response

Response to reviewer comments

We thank the reviewer for all the suggestions and comments to improve the overall quality and suitability of the manuscript for publication. We have addressed all the comments carefully and have highlighted the changes throughout the manuscript.

Reviewer 2

Comments and Suggestions for Authors

Comment 1: Introduction, Lines 63-64, Please add a sentence to emphasize the use of critical control points (eg they could be used to eliminate microbial hazards to an acceptable level etc).

Response: An extra statement was made on lines 66-68 of the revised manuscript to emphasize the purpose of utilizing critical control points in poultry processing.

Comment 2: Introduction, Lines 69-75,

Please describe the main routes of meat contamination with Salmonella (eg intestinal tract of chicken and infection from environment (feed, water, human interventions etc).

Please provide a short description of salmonellosis (eg clinical symptoms)

Response: Additional statements were made to include a brief description of the route for meat contamination in poultry as well as a description of symptoms related to salmonellosis. Please see lines 73-82 in the revised manuscript.

Comment 3: Introduction, Lines 83-95

Same as above for Campylobacter and campylobacteriosis

Response: As suggested, additional statements were included in lines 94 - 98 of the revised manuscript to explain the route of contamination in broiler meat as well as common symptoms of campylobacter infections in humans.

Comment 4: Section 4.4 Cold Plasma Treatment, Lines 562-564

Please provide more information about the mechanism involved (eg conductive electrodes, gas gap, generation of bioactive particles, UV, charged particles etc)

Response: Additional statements were made in lines 533-537 of the revised manuscript to explain the mode of action of nonthermal plasma and the mechanisms involved in NTP treatments.

Comment 5: Finally, the integrity of the cold chain process during product transportation and selling of final product should also be mentioned in this article, as an additional food safety measure (in order to ensure product quality and sustain public health).

Response: As requested by the reviewer, the integrity of cold transportation of food was mentioned in lines 468-469.  However, we respectfully reject the suggestion to include this as an additional food safety measure in the manuscript as it does not fit with our manuscript’s scope and structure. The suggested topic itself is enough for a separate review manuscript.